# Early Influence of Musical Abilities and Working Memory on Speech Imitation Abilities: Study with Pre-School Children

**DOI:** 10.3390/brainsci8090169

**Published:** 2018-09-01

**Authors:** Markus Christiner, Susanne Maria Reiterer

**Affiliations:** 1Department of Linguistics, Unit for Language Learning and Teaching Research, University of Vienna, 1090 Vienna, Austria; 2Centre for Teacher Education, Unit for Language Learning and Teaching Research, University of Vienna, 1090 Vienna, Austria; susanne.reiterer@univie.ac.at

**Keywords:** phonetic language aptitude, intrinsic singing, singing ability, musical aptitude, working memory

## Abstract

Musical aptitude and language talent are highly intertwined when it comes to phonetic language ability. Research on pre-school children’s musical abilities and foreign language abilities are rare but give further insights into the relationship between language and musical aptitude. We tested pre-school children’s abilities to imitate unknown languages, to remember strings of digits, to sing, to discriminate musical statements and their intrinsic (spontaneous) singing behavior (“singing-lovers versus singing nerds”). The findings revealed that having an ear for music is linked to phonetic language abilities. The results of this investigation show that a working memory capacity and phonetic aptitude are linked to high musical perception and production ability already at around the age of 5. This suggests that music and (foreign) language learning capacity may be linked from childhood on. Furthermore, the findings put emphasis on the possibility that early developed abilities may be responsible for individual differences in both linguistic and musical performances.

## 1. Introduction

Musical abilities and the link to language functions have gained considerable scientific interest in the past decade. Music and language are highly intertwined, but despite their similarity remarkably different in many respects. Music, song, and language are all to a large degree acoustic and sensory-motor phenomena, perceived and executed similarly, which might be one of the reasons why investigations have started to compare the three faculties intensively [1,2,3,4]. In language research, understanding positive transfer effects from music to language, which might be induced by musical input/training or may stem from enhanced musical abilities/aptitude, has been of remarkable interest [5,6,7,8,9,10]. Interdisciplinary research comparing and trying to account for the differences and commonalities between music, song and language functions ranges from brain to behavioral and evolutionary to ethological research [2,11,12,13,14]. Comparing musical abilities with language functions often focuses on testing foreign language learning rather than on first language acquisition [15,16,17]. This allows us to observe individual differences more effectively especially when it comes to the link between music and foreign language learning by analyzing the acoustic levels of speech, such as phonetics and pronunciation. New language material, which is unfamiliar to and imitated by the participants informs about individual differences in pronunciation performances illustrating how fast and accurately individuals can adapt to new languages which they have not been exposed to [18,19,20]. Foreign language learning capacity is also influenced by musical training and musicians seem to detect speech incongruities much faster and more accurately than non-musicians [21]. Furthermore, musical training partly influences novel speech processing and learning [7,8,9,22].

Even though language experts have increasingly provided more evidence for individual differences among native speakers’ language proficiency [23,24], inter-learner variation in phonetic abilities is more difficult to observe within the mother tongue compared to other domains like grammar or vocabulary knowledge. In this research we use the term “phonetic language abilities”, “phonetic aptitude” or “speech imitation ability”, interchangeably and what we mean is the capacity to imitate, mimic and pronounce spoken speech based on holistic judgments of human native speaker raters, judging imitated prosody as well as phonetic (segmental) aspects [18]. Referring to aptitude as a more stable “trait” demands developing tasks which are untrained or testing requires minimizing educational/training/experiential influence. This is achieved best by choosing participants who lack experience in what they are tested for. In language aptitude research, pre-school children tested in foreign language capacity and musical abilities are ideal participants because they fulfill both above mentioned criteria. Interdisciplinary research on language aptitude and musical abilities has mostly focused on adults and pupils [3,4,15,16]. Research on pre-school children has largely been neglected so far, despite the fact, that the latter is informative in terms of (phonetic and musical) aptitude, since younger children are still less influenced by education and training (environmental/social influences). Education might be one of the important driving forces supporting children’s progress in cognitive abilities, linguistic development and musical abilities which in turn are related to the social environment in which children grow up. The input children receive correlates to some degree with the output produced [25,26] suggesting that the less formal educational input children receive, the more other factors than training might impact on their performance in foreign languages and music. Even though individual differences will also depend on the input given by caretakers and parents, pre-school children may be most naïve in terms of educational influence compared to older participants and rely on their aptitude while solving problems or learning new skills. In psychology aptitude has been described as a raw material allowing individuals acquire new abilities or adapting behavior faster and more accurately than their peers [27,28]. Aptitude is often considered to be a domain-specific skill and individuals with particular aptitudes show genuine, exceptional and outstanding abilities compared to the general population [29]. This suggests that people who demonstrate certain aptitudes might have at least the potential for outperforming their less talented peers. Research on aptitude is diverse ranging from giftedness with sports, playing chess, composition or writing, to name but a few. Language aptitude has been studied behaviorally [3,4,15,16,28] and less intensively neuro-physiologically, but recent research accumulates knowledge about individual differences in brain structure and function on different levels of linguistic expertise for phonetics and grammar learning [30,31,32,33,34]. More general studies on the bilingual brain investigating polyglots, multi-linguals and bilinguals have revealed evidence of individual differences in language learning and brain processing based on working memory capacity, intelligence, different musical abilities, language exposure and age of acquisition [3,4,18,19,20,35,36]. Similar to language aptitude studies well-known cases of exceptional musicians such as Mozart or Bach do not leave any doubt about the role of aptitude in individual differences in abilities. First evidence that aptitude may also be gene-related and contributes to individual differences in either language or musical abilities has been provided recently. Genetic differences in the auditory pathway have been found to be responsible for differences in music perception [37]. Another longitudinal study has detected that the basic forms and shapes of Heschl’s gyri, which are seen as markers for high musical aptitude in brain research, do not change over time [5,38] and differences in brain structure of musicians have been reported in multiple investigations [39,40,41]. Musicians’ auditory stimulation evokes enhanced activity in a number of originally non-auditory regions in musicians’ brains, such as the sensorimotor, the parietal, the dorsolateral prefrontal cortex, as well as pre-motorand supplementary motor areas [42,43,44,45,46,47,48]. Furthermore, musical training induces plastic changes and influences the complexity in white matter architecture of the cortico-spinal tract [49] and the arcuate fasciculus [50]. Generally speaking, individual differences in musical abilities are said to be based on both nature and nurture related influence. The earlier infants find themselves in a music-rich environment, the better their musical abilities may develop [51]. Regarding nurture related effects it has been reported that musical training during childhood has a significant effect on motor and auditory skills and may lead to structural brain differences in a relatively short period of time [52]. Music training can explain structural brain differences in adult musicians [52] but also directly improves speech segmentation [7], duration perception [6] and pitch perception ability in children [9]. This suggests that both linguistic and musical skills are based on shared neural mechanisms [2,17]. In this interdisciplinary context, there is a rarely mentioned analogy to language acquisition processes. Infants can undoubtedly learn virtually all languages and, for instance, growing up bilingually leads to similar language skills in two languages which they learn without difficulty as a matter of environmental contact [7]. It therefore is an accepted notion that the earlier someone is exposed to a language, the better the achievement [53,54]. Exactly the same seems to be the case for music acquisition processes, even though most investigations that compared music and language focused on different analyses and did not directly compare the acquisition processes per se.

Music and speech are recognized as separate capacities but are perceived by the same auditory system requiring similar cognitive skills [1,2,5]. Achievement in musical abilities improves working memory (WM) capacity, which is again important for multiple cognitive abilities (being neither language- nor music-specific), found to be trainable with transfer effects to general intelligence, executive control and problem solving [55]. WM capacity and its link to musical aptitude has been observed among adults and children [56] and suggestions that verbal and tonal processing and execution show large overlaps (plus subtle differences as a function of musical expertise) have already been shown in a series of neurocognitive investigations e.g., [57,58,59,60]. In addition, research on WM training programs has noted remarkable improvements after training sessions with children suffering from attention deficits, not only in what was trained, but also in new unrelated tasks [61]. WM ability is age-related showing that 3 digits of strings of numbers in a forward order are recalled at around five years of age [62]. Following language aptitude research, it has often been argued that WM has some potential to replace the idea of aptitude and indeed many investigations have been able to detect that WM capacity is related to processing, retaining, and repeating unfamiliar language material [18,19,20,63], placing WM amongst one of the strongest predictors of linguistic success. As already mentioned, WM is age-related, following developmental steps from simpler to more complex. Likewise, this could be similar when learning a new language. In previous research on adults it has been illustrated that 9 to 11 syllable long language material allows us to observe individual differences [18,19]. For pre-school children it can be suggested that 5 to 6 syllable long unfamiliar language material might be appropriate to test their phonetic aptitude.

Apart from WM overlaps of music and speech, song represents a transitional or hybrid faculty, which comprises both linguistic and musical features. Studies focusing on singing capacity and language functions are still underrepresented in recent literature. Some investigations focused on comparing language learning and singing ability [18] and singing as a learning tool [4,64]. Research has also demonstrated the effects of vocal long-term training [50] and found an improved connectivity between the kinesthetic and auditory feedback system and the anterior insular cortex [65], which also contributes to voice motor/somatosensory control and expertise in singers [66]. Furthermore, structural adaptations in singers lead to changes in the complexity and volume of white matter tracts [50].

Comparisons between speech and song [67] concluded that vocalization of speech and song largely shares the same neural network and bilateral activation in the superior temporal sulcus, the inferior pre- and post-central gyrus and the superior temporal gyrus. Speaking and singing draw on common grounds, as body posture, emission, resonance or articulation are based on the same principles [68]. Singing compared to speech is slower in production and trains the motor ability and the vocal apparatus. This is one fundamental reason why singing (intoned word production) is often used for therapeutic purposes to regain motor ability to vocalize and indeed children’s language progress develops alongside motor control [69,70]. 

This investigation focuses on aptitude for acquiring phonetic patterns of unfamiliar languages and its relationship to musical abilities. We sought to uncover the link between phonetic aptitude, singing and musical abilities in pre-school children to better understand music and language acquisition processes from a developmental perspective. We hypothesized that if pre-school children already performed differently in music, singing and phonetic language tasks, like adults do when tested behaviorally, it would be evidence that language or speech imitation aptitude is either developed very early or at least a very stable trait. This could open new discussions and accumulate evidence of the distribution of language aptitude within the general population and thus suggest how aptitude and individual differences can be detected, used and integrated in learning settings to support language acquisition processes. Understanding learners’ needs and individual differences in aptitude may eventually change educational programs which could improve language learning as well as positively affect other related cognitive abilities. 35 pre-school children were tested for their musical abilities (music perception and singing), their ability to imitate Turkish, Tagalog, Russian, and Chinese (phonetic aptitude, speech imitation), their WM (digit span) and social- environmental variables, such as the influence of caretakers and caretakers’ musical activities. This was done with a view to ascertain whether foreign language and musical abilities of pre-school children were comparable to what had been found in adults.

## 2. Methods

### 2.1. Participants

In this investigation we selected 35 (16 female and 19 male) pre-school children at the age of 5 to 6 (mean age = 5.66 and *SD* = 0.48). All of them visited a private kindergarten, were German monolinguals and naïve in formal language and musical training, apart from counting numbers in English and simple singing activities like Happy Birthday. None of them grew up bilingually or participated in a specific language program. A questionnaire revealed that neither the participants nor the parents had had contact to the language material (stimuli) which was tested. The parents belonged to a higher socio-economic background and gave informed consent and agreed to the participation of their children in this investigation. The children were also orally asked whether they liked to participate in the study and all happily agreed because they had already been familiarized with the experimenter on several music teaching occasions. They were instructed to stop at any time, if they felt uncomfortable or if they wished to withdraw their consent. The testing frame took place within two weeks and all tests were performed separately at different times within this time window. The experimenter was well integrated into the kindergarten and started work half a year before their testing to make sure that the children knew him well. For analyzing the background information of the children and the parents, a questionnaire had been designed which focused on musical profiling, singing behavior and language contact. 

### 2.2. Speech Imitation

For testing the children’s ability to remember and repeat unfamiliar language material we selected threephrases for each of the 4 different languages which had been taken (Turkish, Tagalog, Russian, and Chinese). The language material was fiveand sixsyllables long. The original phrases had been spoken by native speakers and recorded in a sound proof room. The children were tested in a separate room in the kindergarten where they had to listen to the phrases threetimes before they repeated the language stimuli, which were recorded and rated by sixnative speakers for Tagalog and Turkish, by fourraters for Russian and sevenraters for Chinese. Ratings were performed on a scale between 0 and 10, where 10 was the highest and 0 the lowest score. Native speakers are said to make judgments comparable to those of phonetic experts [53,54] and multiple investigations used the same methodology to analyze individual differences in phonetic abilities [18,19,20,63]. We instructed the raters to immediately rate the overall performance (spontaneous global judgment) as well as to use headphones while rating the files. 

The analysis of the participants’ singing ability was based on two criteria. First of all, one was how well the children sang according to fourmusic teachers who regularly visited the kindergarten and analyzed the children’s singing ability. They had to rate their singing ability on a scale between 0 and 10, where 0 was the lowest and 10 the highest number. This measurement focused on accuracy, intonation, timing and how well the children sang. For the second criterion the same scale was in use for the ratings. The kindergarten teachers were instructed to observe how intuitively the children started singing without having been instructed to do so (intrinsic motivation) over the period of 14 days. This aimed at isolating the children’s inner needs and intrinsic motivation to sing which should reveal whether those who sing without being instructed may perform differently in language and musicality tasks compared to those who show less motivation to sing. Additionally, the parents were asked to estimate how many hours their children were singing during the week as well as to indicate how many hours they were singing with their children and playing a musical instrument.

### 2.3. Music Perception

The music perception abilities of the children were tested by employing the (Primary Measures of Music Audiation) PMMA [71], a test still widely used in research to measure musicality. This test measures the ability to discriminate tonal and rhythmical changes of paired musical statements and has been designed for children from kindergarten to third grade. The test is subdivided into two sections. While the first one analyzes children’s ability to detect tonal changes, the second one focuses on their ability to discriminate rhythmical changes. Even though this test is widely used for measuring musical aptitude, there are some limitations regarding the validity of the test. For instance, studies reported inconsistent results for the two subtests which show deviations from the published norms [72,73]. Another investigation noted that especially the internal reliability of the rhythm subtest should be treated with caution for grade 1 students and kindergarten children [73].

### 2.4. Working Memory

For testing the working memory abilities of the pre-school children we used strings of numbers/digit span [74]. The numbers were recorded and the children had to listen to the numbers and repeat them in the same chronological order. As a familiarization task, two numbers were given in a string for testing whether the children understood their task. The strings of numbers increased in length and the test stopped after the children could not accurately repeat the strings of numbers a second time.

### 2.5. Ethical Approval

All subjects gave their informed consent for inclusion before they participated in the study. The study was conducted in accordance with the Declaration of Helsinki, and the protocol was approved by the Ethics Committee of the University Hospital and the Faculty of Medicine Tübingen, Project identification code 529/2009BO2.

## 3. Results

### 3.1. Descriptives and Correlations

For illustration of the relationships between the individual variables, tables are shown in the following sections. Table 1 contains the descriptive of the variables under consideration. The units are the actual scores of the variables measured. Table 2 shows the correlations of the variables.

### 3.2. Musicality Test PMMA

The PMMA total score was significantly correlated with the working memory test (strings of numbers which were repeated in forward order), *r_s_* = 0.53, *p* (two-tailed) < 0.01, and the singing parameters singing ability, *r_s_* = 0.44, *p* (two-tailed) < 0.01. The PMMA total score was also significantly correlated with all language imitation tasks, Tagalog (*r_s_* = 0.45, *p* (two-tailed) < 0.01), Chinese (*r_s_* = 0.40, *p* (two-tailed) < 0.05), Turkish (*r_s_* = 0.39, *p* (two-tailed) < 0.05), Russian (*r_s_* = 0.37, *p* (two-tailed) < 0.05) and with the overall speech imitation ability which comprisesof all language imitation tasks (*r_s_* = 0.44, *p* (two-tailed) < 0.01).

### 3.3. Speech Imitation (ComprisesTagalog, Chinese, Turkish and Russian)

Speech imitation showed a significant correlation with the PMMA total score, *r_s_* = 0.44, *p* (two-tailed) < 0.01, with the tonal subtest, *r_s_* = 0.38, *p* (two-tailed) < 0.05. Speech imitation also was significantly related to how accurately the children were repeating strings of numbers in forward order (working memory), *r_s_* = 0.56, *p* (two-tailed) < 0.01 and the singing parameter “singing behavior” which revealed how intuitively the children sang without being instructed to sing, *r_s_* = 0.39, *p* (two-tailed) < 0.05.

### 3.4. Working Memory

The working memory ability correlated with the musicality test PMMA, the total score, *r_s_* = 0.53, *p* (two-tailed) < 0.01, the tonal subtest, *r_s_* = 0.58, *p* (two-tailed) < 0.01. Further correlations of the working memory capacity to all language imitation tasks, Tagalog (*r_s_* = 0.42, *p* (two-tailed) < 0.05), Chinese (*r_s_* = 0.41, *p* (two-tailed) < 0.05), Turkish (*r_s_* = 0.58, *p* (two-tailed) < 0.01), Russian (*r_s_* = 0.38, *p* (two-tailed) < 0.05) and with the overall speech imitation ability (*r_s_* = 0.56, *p* (two-tailed) < 0.01) were also observed.

### 3.5. Singing Ability

Singing ability, which measures how accurately the children sang, was correlated with the PMMA total score (*r_s_* = 0.44, *p* (two-tailed) < 0.01), the tonal PMMA subtest (*r_s_* = 0.39, *p* (two-tailed) < 0.05) and the rhythm PMMA subtest (*r_s_* = 0.34, *p* (two-tailed) < 0.05). Singing ability also showed a significant relationship to singing behavior (*r_s_* = 0.80, *p* (two-tailed) < 0.01) and singing ability also correlated with the language imitation task Tagalog (*r_s_* = 0.44, *p* (two-tailed) < 0.01).

### 3.6. Singing Behavior

Singing behavior which should reveal the intrinsic motivation of the children to sing, correlated with the rhythm PMMA subtest (*r_s_* = 0.36, *p* (two-tailed) < 0.05) and singing ability, *r_s_* = 0.80, *p* (two-tailed) < 0.01 and showed a significant relationship to Tagalog *r_s_* = 0.49, *p* (two-tailed) < 0.01, and to the global speech imitation ability, *r_s_* = 0.36, *p* (two-tailed) < 0.05 as well.

### 3.7. Inter-Rater Reliability

The inter-rater reliability Cronbach’s alpha was applied and was 0.88 for Tagalog, 0.85 for Chinese, 0.94 for Turkish and 0.86 for Russian which are all in the acceptable range above 0.70.

### 3.8. Whitney–Mann Test (Group Comparisons)

We divided our group into high and low musical aptitude based on the music perception task PMMA (tonal and rhythmic discrimination ability). The high musical aptitude group (*Median* = 4) performed significantly better than the low musical aptitude group (*Median* = 3) in the working memory task, U = 51.00, z = −3.58, *p* < 0.001, r = −0.61. The high musical aptitude group (*Median* = 3.25) also performed significantly better than the low musical aptitude group (*Median* = 2.75) in speech imitation, U = 83.00, z = −2.28, *p* < 0.005, r = −0.39. The high musical aptitude group (*Median* = 7.88) performed significantly better than the low musical aptitude group (*Median* = 4) in singing (singing ability), U = 91.50, z = −2.01, *p* < 0.005, r = −0.34. The significance of all three group comparisons was also given after Bonferroni–Holm-Correction. Significance was inferred at *p* < 0.05 after Bonferroni–Holm-Correction for the three variables working memory, singing ability and speech imitation. For illustration see Figure 1 below.

## 4. Discussion

The music perception task (PMMA) was used to create group membership of the children and consists of rhythm and tonal discrimination tasks. Even though this test is widely used for measuring musical aptitude, there are some limitations concerning the validity of the test. According to Stamouet et al. [73], the results of the rhythm subtest should be regarded with caution for pre-school children and grade 1 students as a result of cross-cultural issues which may also be relevant for this investigation as the children were German native speakers. However, in this study similarly to research on adults, effects of individual differences in music perception, working memory capacity, speech imitation and singing have been found. For statistical analysis we split the groups based on music perception abilities, measured by the PMMA and we created two groups of high and low musical aptitude. Working memory, speech imitation and singing ability have been found to be significantly different between the two musical groups created, while singing behavior has failed to reach statistical significance within the model, even though there was a tendency.

### 4.1. Musical Expertise, Plasticity, Musical Abilities and Working Memory

Several investigations have found that music perception abilities improve the ability to remember, imitate and retrieve unfamiliar language material [5,15]. As shown in this investigation, the same seems to hold true for pre-school children: The higher their music perception ability, the better their speech imitation ability to memorize and imitate new, unfamiliar language material. This puts emphasis on the importance of addressing the relationship between musical and linguistic abilities. The building of expertise in the areas of linguistic or musical abilities is, in essence, the recurrent problem of the relationship between “nature and nurture”, difficult to investigate experimentally, given a lack of testable definitions of the “talent-ability” terms. There is considerable debate as to whether differences in behavior are due either to “innate talent” or to the quantity and quality of practice in a given domain. Indeed, the viewpoint of interaction between genetically and epi-genetically driven abilities modified by experience is perhaps dependent on the skill domain, but in general widely accepted in the domains of language or music acquisition.

Musicians’ neurophysiological, auditory enhancements and beneficial transfer effects on language functions are often related to the years of training [50,52,75,76,77,78], and ontogenetically speaking, discrimination of rhythm is an early developed mechanism that starts prenatally and continues during infancy [77]. The effect of musical training and the impact of culture on music acquisition is undeniable, even though individual differences in high achievement may be based on other aspects as well. In this study the children were naïve in terms of individual musical training which might suggest that other factors than proper music training, such as very early developmental or pre-, peri- or post-natal influencescontribute to individual differences in their musical abilities. Auditory models have already proposed that primary capacities influence musical aptitude, while secondary musical skills are environmentally shaped by the culture and individual training received [79]. First evidence that primary capacities for musical aptitude may also be gene-related, like processing of auditory signals, which alter the auditory pathway crucial for discriminating musical input, has already been reported [37]. Multigenerational family studies have also revealed that several predisposing genes or variants contribute to musical aptitude e.g., [80] and evidence for alterations of the brain structure of musicians, which improve music perception and performance, are diverse e.g., [5,20,40,49,51]. Inter-individual changes and structural differences in the auditory cortex cannot be ascribed to training effects only and particular brain areas, seen as markers for high musical abilities (e.g., Heschl’s gyrus) seem to be rather stable in shape [5,38,40,41]. For instance, duplications of Heschl’s gyrus occur more often in musicians rather than in non-musicians [38]. Anatomical alterations of the gray matter [33,34] or volume and complexity differences in white matter tracts of singers [50] have been reported. While increasingly more evidence manifests that musical aptitude could be gene-related or at least a stable trait over life-time, studies that identify biological markers, such as genetic or neuroanatomical markers for phonetic aptitude, have been largely neglected so far within the area of second language acquisition or language learning research.

The underlying reasons for the link between musical and phonetic aptitude are also based on shared cognitive functions as well as on shared mechanisms in the execution and processing of music and speech [5,6,7,8,9,10]. Music and language functions require the recruitment of similar cognitive processes, attention control, anatomical and neuroanatomical endowment [5]. WM capacity, an elaborate cognitive skill, crucial for multiple abilities, is related to phonetic aptitude and musical abilities. The ability to remember rapid and temporary information is important for the learning of new language material which is poor in linguistic content [81]. There is an analogy to remembering melodies or discriminating different musical pieces. Language learners in the beginning phase will benefitmost from higher WM capacity. The basic acoustic signals of musical sounds (pitch, timing and timbre) play a key role in both speech and music, especially in conveying information [82]. Pitch, the property of sounds that is organized by a scale, can be judged as lower or higher. Timing refers to temporal events of sounds and timbre to the perceived sound quality also referred to as tone quality [82]. Evidence has been provided that the improved processing of timbre and pitch in musicians is based on functional adaptations, but seems to play a general role in musical development from infancy onwards [83]. Furthermore, musicians’ improvements in detecting pitch and timbre cues largely rely on plastic adaptations [49,82,84]. Language acquisition processes are also based on differentiating timbres of speech that are meaningful and/or meaningless [84] and for musicians it is also necessary to discriminate timbre differences between various instruments which in turn seem to have an effect on speech sound discrimination as well [82]. Evidence for an overlap between processing tonal and verbal material, especially comes from brain research e.g., [51,53]. Higher WM abilities are also associated with musical aptitude [18], but are also age related [85]. Children (aged 4 to 6) can remember around twowords in a forward order, and threedigits of strings of numbers in a forward order [62]. The results of this as well as earlier research corroborate the findings of previous investigations and suggest that WM is highly important for learning new languages. Thus, it can be assumed that people with high WM capacity are faster learners [86], showing that WM predicts not only overall language aptitude scores, but is related to the language analysis components of the aptitude construct. Therefore, the link between WM, language acquisition and musical aptitude is a very promising research area and the overlaps of WM capacity for music and language functions may lead or has led researchers to argue that WM has the potential to replace the whole language aptitude construct (equaling language aptitude with working memory capacity). This claim, however, should be treated with caution, since one limitation of direct imitation tasks, like used in this investigation, is that it always requires high working memory loads. Indirect imitation tasks retrieved from long-term memory may reduce WM influence and inform about phonemic awareness. Future research on phonetic aptitude should include both direct and indirect imitation tasks to get a multidimensional impression of phonetic language aptitude. The measurements used in this investigation provided evidence on how fast and accurately someone imitates, retrieves and memorizes unfamiliar speech material on an “ad-hoc” basis, while indirect imitation tasks would also yield information about achievements in meta-cognitive awareness (phonemic awareness).

### 4.2. Singing Behavior, Singing Ability and Speech Imitation

Singing behavior was not significantly different in the music groups created which is in line with previous research where music perception ability of singers and instrumentalists did not differ, while their ability to reproduce new language material was significantly better in the singer group [19]. The result of this investigation, however, could also be due to the fact that pre-school children do not have comprehensive musical skills. Even though sensory consonance and pitch discrimination ability develops relatively early during infancy, harmonic knowledge, for instance, develops significantly later between 6 and 12 years [87]. Furthermore, limitations of the study design must be mentioned here as well. The ratings of the singing behavior were rated based on mere observations by the caretakers in the kindergarten. Our reasons for choosing this design lay in keeping testing time appropriately short for pre-school children and in rendering the testing situation as natural as possible.

Individual differences in the performances of children were also based on intrinsic motivation to produce vocalizations, which might be driven by their inner needs to express their feelings to the outer world. The variable singing behavior correlated more with to the global speech imitation score than mere singing ability or singing accuracy. The reason for this might be that children’s fine motor abilities are still under development and this affects both, the singing and the articulatory skills [3,4]. First language acquisition develops along motor control development [88] and singing as vocal behavior largely shares the same mechanisms. Evidence that vocal motor commands can alter and influence speech perception has already been provided e.g., [89]. Intrinsic singing behavior and the desire to sing, reflecting intrinsic motivation to sing, are maybe more important than accuracy to playfully expand vocal flexibility in this period of life. However, since correlations differed only between one another in the case of the single contributions of singing ability and singing behavior, these results should be interpreted with caution.

### 4.3. Typologically Different Languages and Musical Abilities

Tone languages and non-tone languages are different in many respects and even though causality cannot be explained, based on the correlations, there seems to be a tendency towards differences in non-tone languages and tone-languages in relation to singing. Future investigations may need to consider typologically different languages and their relationship to musical aspects which cannot be explained within this limited research design. Recent research has isolated different transfer effects from music to language, where pitch discrimination contributed to tone-languages, while rhythmic discrimination contributed to non-tone languages [63]. Goswami and colleagues also showed that “novel remediation strategies on the basis of rhythm and music may offer benefits for phonological and linguistic development” [90] and early musical training during childhood supports foreign language perception, memory and later foreign language acquisition processes [91]. As a general rule, it has been accepted that during infancy, basically between six and twelve months, language specific phonetic contrasts are perceived, followed by a decline in the ability to perceive non-native contrasts [92]. The first twoyears of development, therefore, seem to leave a deep cognitive imprint on children’s language performances also later in life [93]. Twelve months old infants’ speech segmentation and speech processing abilities seem to be predictive measurements for observing individual differences in language development between fourand 6 years of age [94]. The children in this investigation were five-to six-year-olds and had already acquired native phonetic contrasts for the German language, a non-tone language. Singing ability and singing behavior did not seem to relate to the Chinese performances. Chinese only correlated with the PMMA total and tonal score and working memory capacity, but did not show correlations to the rhythm PMMA subtest, or to any of the singing criteria. Although correlations do not inform about causality, further research on tone language imitation and its relation to musical measurements should be investigated in more detail. This could include various musical measurements focusing on pitch, timing and timbre perception tasks, musical instrument playing or singing to better understand the impact of musical abilities on second language learning of non-tone language speakers.

An analysis of Chinese imitation performances of school children at the age of 9 have revealed that around 40% of the variance of Chinese imitation of non-tone (German) language native speakers could be explained by singing ability, tonal perception ability together with WM capacity [63]. Learning Chinese as a second language may require precise musical knowledge an ability which is developed at the age of 9 [95]. Research has shown that tone language speakers are better at discriminating tone contrasts in any language compared to non-tone language speakers [96] and evidence that musicians and tone-language speakers share cognitive and perceptual skills for pitch discrimination has also been reported recently [97]. Musicians and tone-language speakers largely share cognitive and perceptual skills for pitch discrimination which illustrates a bidirectional path between music and language [97]. Evidence has been provided that high melodic ability of non-tone language speakers leads to better performances in detecting tonal variations in Mandarin [98]. This shows that musical ability may be highly relevant for learning tone-languages as a non-tone language speaker. This may be the reason as to why Chinese native speakers who learn a non-tone language during adulthood face pronunciation difficulties, and vice versa, non-tone language speakers have difficulties to discriminate Chinese sounds. Singing should be more closely integrated in future research designs to isolate the influence of singing on the acquisition of tone languages as a second language of non-tone language speakers.

## 5. Conclusions

Musical ear, singing ability/behavior and working memory capacity are linked to speech imitation abilities already at a very early stage in development. Comparable to research on adults, we have found similar effects and links in pre-school children, who are naïve in terms of education and music training, suggesting that individual differences might also be based on very early developed factors. Group comparisons of children with high versus low music perception abilities reveal, that the high musicality groups perform better in novel speech imitation, can sing better and show enhanced WM capacity. Children at this particular age are less influenced by educational input than adults, which hints at early developmental factors contributing to individual differences in musical abilities and (novel) speech learning abilities. This shows that music and language capacities are ultimately linked in children and adults. Singing behavior did not yield statistically significant differences between groups, which could show that this behavioral measure displayed lower reliability. On the other hand, it is in line with research on adults that singing ability, singing motivation and music perception are different, non-overlapping entities, sometimes leading to similar and sometimes to different transfer effects [19].

## Figures and Tables

**Figure 1 brainsci-08-00169-f001:**
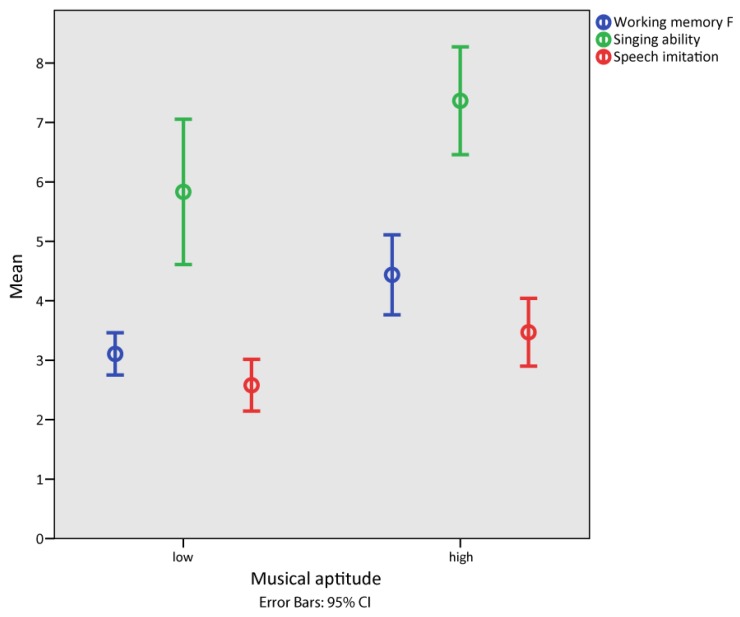
Working memory, singing ability and speech imitation was higher in children with high (compared to low) musical aptitude.

**Table 1 brainsci-08-00169-t001:** The descriptive of the variables under consideration.

Descriptive				
Variables	*M*	*SD*	*Min*	*Max*
Working memory forward	3.71	1.20	2.00	8.00
PMMA total	52.94	6.93	38.00	65.00
PMMA tonal	28.34	4.46	18.00	36.00
PMMA rhythm	24.60	3.89	17.00	34.00
Singing ability	6.53	2.30	0.25	10.00
Singing behavior	5.35	2.39	0.25	10.00
Speech imitation	2.99	1.07	0.87	5.80
Tagalog Mean	3.84	1.15	1.06	7.17
Chinese Mean	2.77	0.84	1.09	4.23
Turkish Mean	3.01	1.76	0.33	7.06
Russian Mean	2.32	1.33	0.17	4.92

PMMA: Primary Measures of Music Audiation.

**Table 2 brainsci-08-00169-t002:** Correlations between the individual variables.

Correlations (Spearman)											
Variables	PMMA Total	PMMA Tonal	PMMA Rhythm	Working Memory F	Singing Ability	Singing Behavior	Speech Imitation	Tagalog Mean	Chinese Mean	Turkish Mean	Russian Mean
PMMA total	1	0.84 **^†^	0.77 **^†^	0.53 **^†^	0.44 **^†^	0.32	0.44 **^†^	0.45 **^†^	0.40 *^†^	0.39 *^†^	0.37 *
PMMA tonal	0.84 **^†^	1	0.35 *	0.58 **^†^	0.39 *^†^	0.23	0.38 *^†^	0.40 *^†^	0.42 *^†^	0.34 *	0.28
PMMA rhythm	0.77 **^†^	0.35 *	1	0.26	0.34 *	0.36 *	0.31	0.35 *	0.19	0.28	0.29
Working memory F	0.53 **^†^	0.58 **^†^	0.26	1	0.32	0.29	0.56 **^†^	0.42 *^†^	0.41 *^†^	0.58 **^†^	0.38 *^†^
Singing ability	0.44 **^†^	0.39 *^†^	0.34 *	0.32	1	0.80 **^†^	0.25	0.44 **^†^	−0.04	0.21	0.23
Singing behavior	0.32	0.23	0.36 *	0.29	0.80 **^†^	1	0.39 *^†^	0.53 **^†^	−0.09	0.36 *	0.38 *^†^
Speech imitation	0.44 **^†^	0.38 *^†^	0.31	0.56 **^†^	0.25	0.39 *^†^	1	0.69 **^†^	0.61 **^†^	0.90 **^†^	0.89 **^†^
Tagalog Mean	0.45 **^†^	0.40 *^†^	0.35 *	0.42 *^†^	0.44 **^†^	0.53 **^†^	0.69 **^†^	1	0.23	0.57 **^†^	0.53 **^†^
Chinese Mean	0.40 *^†^	0.42 *^†^	0.19	0.41 *^†^	−0.04	−0.09	0.61 **^†^	0.23	1	0.56 **^†^	0.47 **^†^
Turkish Mean	0.39 *^†^	0.34 *	0.28	0.58 **^†^	0.21	0.36 *	0.90 **^†^	0.57 **^†^	0.56 **^†^	1	0.72 **^†^
Russian Mean	0.37 *	0.28	0.29	0.38 *^†^	0.23	0.38 *^†^	0.89 **^†^	0.53 **^†^	0.47 **^†^	0.72 **^†^	1

** Correlation is significant at the 0.01 level (2-tailed). * Correlation is significant at the 0.05 level (2-tailed). ^†^ Correlation is significant after Benjamini-Hochberg correction for overall false discovery rate *p* ≤ 0.05. PMMA: Primary Measures of Music Audiation.

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
