# Peer review of "Early Influence of Musical Abilities and Working Memory on Speech Imitation Abilities: Study with Pre-School Children"

_brainsci, 2018, doi:10.3390/brainsci8090169_

Round 1
Reviewer 1 Report
The paper has been very thoroughly reviewed and all relevant issues raised in the previous version have been adequately addressed.
Author Response
Dear Reviewer,
we say thank you for your comments. We tried to correct some errors we came across
thank you
Reviewer 2 Report
This paper is an interesting though modest contribution, in particular by focusing on pre-school children and their benefits from singing. The article is well written and the authors show some knowledge of the field. The paper as a whole, however, is not totally convinging and could benefit from some modification in order to improve its overall quality. I list below some general remarks and some more detailed comments.
General remarks
The theoretical and background is rather limited. Much more can be said about musical development, ability in general, and musical ability in particular, and also about the domain of neuroplasticity as applied to music. In line with this, the list of references could be expanded considerably. There is actually mostly one reference for one claim, but claims can be made much stronger by providing an overview of the relevant and most recent literature. By providing always a limited set of references (mostly only one) there seems to be some gratuitousness in selecting this reference. Some references are also not very strong (proceedings, poster session).
The methodology used is not always very strong. This holds in particular for the ratings which seem to be rather subjective and without much rigor.
The ecological validity of the testing of musical ability can be questioned. Gordon’s test is rather narrowly defined. It tests only tonal and rhythmic discrimination ability, but what about timbre, which is so important in the learning of the phonetics of language. Think for instance of the role of formants, which are not even mentioned in this paper.
The figure captions and table legends could be explained better. What about the units of the y-axes? What do they stand for?
The domain of neuroplasticity is not explained in depth ( see e.g. the contributions by Reybrouck, Brattico, Vuust, Koelsch, and many others). This is a very hot topic with lots of contributions, which are not mentioned in this paper. There is also much more than just Heschl’s gyrus. This section is not at academic standards.
Some claims are somewhat gratuitous.
Detailed comments
- line 35: please add at least some references to give a first overview of the theoretical and empirical background of these claims
- line 40: the sentence is not clear and is difficult to understand, please explain better
- line 62: the topic of aptitude in psychology is not dealt with in depth. Please provide a more substantial approach and add some more references.
- line 105: very few references here; there is much more material to be found
- line 120: difficult to understand; explain better the concept of ethnology of language aptitude
- line 147: some minor spelling mistakes: add blank space between “the4different”
- line 168: I have doubts about the ecological validity of the PMMA.
- lines 158-177: these matters are treated in a very superficial way, not even at academic standards. Provide much more substantial background, definitions and references.
- line 185 ff: provide figure captions and table legends. Explain the units used. How were they measured? What are the scales, the ranges of the score?
- line 188: what does the F stand for after Working memory? Please explain. The table and the table legend should be self-sufficient so that readers can understand them even without reading the main text.
- line 189: please explain the correlations better in the main text
- line 194: what about timbre?
- line 221-225: please explain the units in the y-axis
- line 230: please discuss a little the limited ecological validity of PMMA
- line 251: delete comma after like
- line 251: the contents of this sentence are questionable. There is more than just the genetic factor. See the abundant literature on music and neuroplasticity and the role of nurture as against nature; see also the role of ontogenetic development
- line 257: there is so much more than Heschl’s gyrus; very limited approach to the matter here; also very few references
- line 271: I miss some discussion of the role of timbre here
- line 274: minor spelling mistakes: add blank space between “and3digits”
- line 300: what is meant with “harmonic knowledge”; to wat extent is this essential for singing accuracy? these claims are somewhat gratuitous
- line 320: delete blank space between tone- and languages
- line 330: this is not clear and difficult to understand; please explain better
- line 337: same remark
Author Response
Dear reviewer,
we want to say thank you for your valuable comments and we tried to improve the manuscript and hope to have addressed the issues which were relevant to be changed. We attached the changes made and your comments. We hope that this is fine for you.

Round 2
Reviewer 2 Report
I had a look at the revision of the paper and it seems to me that he paper is OK now.
Author Response
Academic Editor Notes for major revisions
Academic Editor Notes
We want to thank you for your valuable comments. We changed large parts of the manuscript and hope that we addressed the issues accordingly
Best wishes
Markus Christiner and Susanne Maria Reiterer
Abstract:
1) Lines 13-14. It is unclear how research on pre-school children’s musical and language abilities should provide insights into the origin of language and music-related aptitudes. I would rather suggest emphasizing the relationships between the two domains.
See lines :
Musical aptitude and language talent are highly intertwined when it comes to phonetic language ability. Research on pre-school children’s musical abilities and foreign language abilitiesare rare,but give further insights about the relationship between language and musical aptitude.
2) Lines 20-23. Please turn down this last statement. There is no causal relationship.
Introduction:
This suggests that music and (foreign) language learning capacity may belinked from childhood on,and puts emphasis on the possibility that early developed or pre-existential abilitiesmay beresponsible for individual differences in both linguistic and musical performances.
3) Lines 27-32. I miss adequate references.
See references [1-4].
4) Lines 31-34. The authors should cite papers that addressed causal and correlative relationships between music training and speech processing. Please have a closer look to the papers of Mireille Besson et al., Schellenberg et al., Elmer et al., Francois et al., etc.
See references [5-10].
5) Lines 34-36. This sentence is quite vague and should be specified / differentiated.
Interdisciplinary research on music, song and language functions therefore is diverse, ranging from brain to behavioral, evolutionary to ethological research [11, 2] which aim at isolating shared and distinct mechanisms to better understand their relations and origin
6) Lines 39-42. Please describe and cite previous papers that have addressed relationships between music training and foreign language processing / learning (see for example the seminal study of Patrick Wong).
This allows observing individual differences more effectively,especially when it comes to the analysis of the acoustic levels of speech such as phonetics and pronunciation. New language material, which is unfamiliar to and imitated by the participants, allows observing individual differences in pronunciation performances illustrating how fast and accurate individuals can adapt to new languages which they have not been exposed to [12-14]
7) Lines 54-55. The authors should cite the relevant literature addressing transfer effects from music training to speech processing. To only cite own studies is not at all adequate.
See references [e.g. 3-4, 19-20].
8) Lines 73-75. The authors should cite the seminal papers of Jubin Abutalebi and Daniela Perani on bilingualism / multilingualism as well as the literature on language expertise, including simultaneous interpreters and phoneticians.
Language aptitude has been studied intensively [24, 3-4, 19-20] and studies on polyglots, multilinguals and bilinguals have revealed evidence for individual differences in language learning abilities based on working memory capacity, intelligence,different musical abilities, language exposure and age of acquisition [26-27, 12-14, 3-4].
9) Lines 80-83. The authors should more precisely address the functional and anatomical plasticity effects that have repeatedly been reported in musicians, including the planum temporale, the motor/somatosensory cortices, the parietal lobe, and the ventral part of the prefrontal cortex. I also miss a description of the fiber tracts that are classically altered in musicians, namely the corticospinal tract, the arcuate fasciculus and the corpus callosum.
Another longitudinal study detected that the basic forms and shapes of Heschl’s gyri, which are seen as markers for high musical aptitude in brain research, do not change over time [5, 29] and differences in brain structure for musicians have been reported in multiple investigations [30-32].Musicians auditory stimulation evokes enhanced activity in a number of originally non-auditory regions in musicians’ brains, such as the sensorimotor, the parietal, the dorsolateral prefrontal cortex, as well as premotor (PMC) and supplementary motor areas [33-39] as well as musical training induces plastic changes and influences the complexity in white matter architecture of the corticospinal tract [40] and the arcuate fasciculus [41].
10) Lines 84-85. Please cite longitudinal studies on music training in childhood. See for example the seminal studies of Krista Hyde. Also the studies of the group of Mireille Besson and Mari Tervaniemi should adequately be described and cited.
Infants can undoubtedly virtually learn all languages and, for instance, growing up bilingually leads to similar language skills in two languages which they learn without difficulty as a matter of environmental contact [7].
11) Lines 87-89. Please cite the literature on speech processing in infants. See for example the work of Clément Francois or the papers of the group of Mari Tervaniemi.
Following music experts, individual differences in musical abilities are said to be based on nature and nurture related influence [42].
12) Lines 89-90. Please add references. Please also do that for the next sentence.
13) Lines 93-94. Please add adequate references.
Exactly the same seems to be the case for music acquisition processes, even though most investigations that compared music and language focused on different analyses and did not directly compare the acquisitional processes per se. Music and speech are recognized as separate capacities but are perceived by the same auditory system requiring similar cognitive skills[1, 2, 5].
14) Line 94-97. Please cite the relevant literature pointing to relationships between music training and cognitive functions, including Jennifer Zuk, Vanessa Sluming, Stefan Kölsch, Glen Schellenberg, and others.
See references [47-50].
15) Lines 125-128. Pease add adequate references.
See references [41]. [o1]
Methods:
16) Please specify the number of participants for each gender.
See participants
17) Please add additional information about the sociodemographic status of the parents.
The parents belong to higher socio-economic background and gave informed consent and agreed to the participation of their children in this investigation.
18) Did the authors test the inter-rater reliability? This should be done and reported in the results section.
Inter-rater reliability
The inter-rater reliability Cronbach’s alpha was applied and was .88 for Tagalog, .85 for Chinese, .94 for Turkish and .86 for Russian (seven raters for each language and item) which are all in the acceptable range above .70.
Results:
19) Please indicate which correlations would survive a correction for multiple comparisons.
Please see correlations table we used † Benjamini-Hochberg correction for overall false discovery rate p <= 0.05.
20) Lines 294-295. Please insert citations from other research groups.
References [e.g. 3-5, 19].
21) Lines 297-298. As previously mentioned in the abstract, I don’t agree on the fact that the results of the authors contribute to the debate on the origin of musical and language-related abilities. Please turn down this statement and more likely argue in terms of relationships.
This puts emphasis on the importance of addressing the relationship between musical and linguistic abilities.
22) “Musicians’ neurophysiological and auditory enhancements are often related to the years of training [56]…” Please add more references.
References [41, 66-67],
23) Lines 323-330. Please insert specific references for each statement.
The underlying reasons for the link between musical and phonetic aptitude are also based onshared cognitive functionsas well ason shared mechanisms in theexecution and processing of music and speech [5-10]. Music and language functions require the recruitment of similar cognitive processes, attention control, anatomical and neuroanatomical endowment [5]WM capacity, an elaborate cognitive skill, crucial for multiple abilities, is related to phonetic aptitude and musical abilities. The ability to remember rapid and temporary information is crucial for the learning of new language material which is poor in linguistic content [69].There isan analogy to remembering melodies or discriminating different musical pieces.
L
24) Lines 334-336. Please insert references about the statement that the advantages of musicians in pitch and timbre processing are related to plastic adaptations.
Timing refers to temporal events of sounds and timbre to the perceived sound quality also referred to as tone quality [70]. Evidence has been provided that the improved processing of timbre and pitch in musicians is based on functional adaptations, but seems to play a general role in music development from infancy onwards [71].Furthermore, musicians’ improvements in detecting pitch and timbre cues largely rely on plastic adaptations [40, 70, 72].
25) Lines 393-394. Please insert adequate references about transfer effects from music training to speech processing. Maybe also the novel paper of Eva Dittinger on phonetic processing in musicians and non-musicians (Eur. J. Neurosci.) might be of interest. Have also a look to the recently published study of Yun Nan (PNAS).
Goswami and colleagues also showed that “novel remediation strategies on the basis of rhythm and music may offer benefits for phonological and linguistic development“[78] and early music training during childhood supports foreign language perception, memory and later foreign language acquisition processes [79].
26) Lines 415-418. Please more deeply discuss the literature on music training and tone languages.
Learning Chinese as a second language may require precise musical knowledge or well developed musical skills, an ability which is developed at the age of 9[83]. Research has shown that tone language speakers are better at discriminating tone contrasts in any language compared to non-tone language speakers [84] [o2] and evidence that musicians and tone-language speakers share cognitive and perceptual skills for pitch discrimination has also been reported recently [85]. Musicians and tone-languages speakers largely share cognitive and perceptual skills for pitch discrimination which illustrates a bidirectional path between the music and language [85].Evidence has been provided that high melodic ability of non-tone language speakers leads to better performances in detecting tonal variations in Mandarin [86] showing that musical ability may be highly relevant for learning tone-languages as foreign language as non-tone language speaker.
Figures and tables:
27) I miss a table legend for Table 1.
Table 1 shows the descriptive of the variables under consideration.
28) What is depicted in Figure 1? Please specify more precisely.
Figure 1 shows that high and low music perception ability has a significant effect on working memory, singing ability and speech imitation.
[o1]
[o2]
